# Preliminary Development of Global–Local Balanced Vision Transformer Deep Learning with DNA Barcoding for Automated Identification and Validation of Forensic Sarcosaphagous Flies

**DOI:** 10.3390/insects16050529

**Published:** 2025-05-16

**Authors:** Yixin Ma, Lin Niu, Bo Wang, Dianxin Li, Yanzhu Gao, Shan Ha, Boqing Fan, Yixin Xiong, Bin Cong, Jianhua Chen, Jianqiang Deng

**Affiliations:** 1Hainan Provincial Tropical Forensic Engineering Research Center & Hainan Provincial Academician Workstation (Tropical Forensic Medicine), Key Laboratory of Tropical Translational Medicine of Ministry of Education, School of Basic Medicine and Life Sciences, Hainan Medical University, Haikou 571199, China; xmbebrave@163.com (Y.M.); sapphirestrings@sina.com (B.W.); cjhtin86591@163.com (J.C.); 2School of Biomedical Informatics and Engineering, Hainan Medical University, Haikou 571199, China; linniu@hainmc.edu.cn (L.N.);; 3Key Laboratory of Forensic Medicine, Department of Forensic Medicine, Xinjiang Medical University, Urumqi 830017, China; 4School of Public Health, Hainan Medical University, Haikou 571199, China; 5Department of Forensic Medicine, Hebei Medical University, Shijiazhuang 050011, China

**Keywords:** forensic entomology, Artificial Intelligence, sarcosaphagous flies, species identification, fine-grained image classification

## Abstract

Identifying the species of flies that are active on corpses is important for forensic work, but current methods are complex, require experts to operate, and are difficult for ordinary people to perform. In order to make it easy for everyone to identify these flies, we have in the past developed a new artificial intelligence model to identify fly species. This time, we optimised this model and also made this program into a small app for phones. It can identify common fly species in Hainan with good recognition and high accuracy. If we encounter a few cases that are difficult to identify, we have prepared a supplementary method to determine their species by testing the DNA of the flies. As we continue to improve this programme, it will become more and more applicable to forensic work, improving the efficiency and accuracy of forensic identification and thus better assisting in the detection of cases.

## 1. Introduction

Entomological evidence provides important entry points and clues for the investigation of forensic science cases involving the assessment of post-mortem interval (PMI) as well as place and cause of death. The accurate identification of sarcosaphagous insect species is a primary and vital step in conducting such research. If errors occur during the identification process, it will directly affect the accuracy of the subsequent investigation results, so the relevant professionals are required to accurately identify the species of the relevant insects and their developmental stages at the scene [1]. At present, it is still the gold standard for forensic entomology to identify the species by analysing the morphological characteristics of the insects collected at the scene. However, in forensic science practice, site investigators generally lack basic forensic entomology knowledge. They also do not have rich practical experience in long-term insect classification work and fail to master species identification skills [2]. The existence of these difficulties seriously limits the wide application as well as the role of forensic entomology in forensic science practice and wastes the value of entomological evidence for use in real cases. Compared with morphological methods, molecular biology identification methods require relatively low professional skills and knowledge of insect morphological classification, and after years of data accumulation, the reliability of molecular species identification results based on the sequencing of insect DNA-specific fragments and related insect gene databases has been widely recognised by the industry [3], but it must rely on a complete set of research equipment, space and cumbersome laboratory operations, which is troublesome and costly, and does not achieve the convenient use of forensic entomology.

In recent years, the rapid development of Artificial Intelligence (AI) technology has made it feasible to build automated insect identification and classification systems [4]. The intervention of AI technology provides a practical solution for operators who lack expertise and experience in species identification [5]. At the same time, significant improvements in digital photography and smartphone performance have led to the widespread availability of smartphones equipped with advanced cameras, providing the convenience of using AI for image processing on mobile devices at hand. Computer vision and pattern recognition are key to AI technology, which has been widely used in image classification for production safety management, healthcare, and the identification of plants and animals [6,7,8,9]. The development of related applications also provides new tools for rapid species identification. Researchers have used AI and related derived technologies to identify insects to assist in pest and disease identification and management [5,10,11]. However, given the complexity of insect species, there is a lack of AI that can be used to identify all insects, and targeted exploratory research must be carried out to suit the needs of this research area. Currently, there are fewer studies on sarcosaphagous insect AI for forensic entomology applications, and there is an urgent need for research in this area.

Deep learning is currently a commonly used technical idea in AI model creation, and convolutional neural networks (CNNs) have shown superior accuracy to traditional image processing methods in image classification tasks [12]. Current methods based on convolutional neural networks focus on utilising local information, whereas back-transformer-based methods emphasise on extracting global information. However, in the image classification of flies, both methods go beyond the balance between global and local information, resulting in an inadequate representation of non-granular fly characteristics, which seriously affects the accuracy of insect classification. In our previous research, we innovatively proposed the GLB-ViT network architecture, cleverly integrated the GLB-local feature extraction mechanism, and established a preliminary method system. GLB-ViT is designed to classify forensic flies by integrating global and local feature extraction. crucial for distinguishing subtle differences in fly species. Its feature fusion module optimally integrates these features, enhancing accuracy. The model’s lightweight design reduces computational costs while maintaining efficiency, and its robustness handles diverse environmental conditions. The database for this model initially includes images of six species: *Sarcophaga dux* (Thomson, 1869) (Diptera: Sarcophagidae), *Sarcophaga sericea* (Walker, 1852) (Diptera: Sarcophagidae), *Sarcophaga misera* (Walker, 1849) (Diptera: Sarcophagidae), *Sarcophaga peregrina* (Robineau-Desvoidy, 1830) (Diptera: Sarcophagidae), *Chrysomya pinguis* (Walker, 1858) (Diptera: Calliphoridae), *Lucilia cuprina* (Wiedemann, 1830) (Diptera: Calliphoridae). The model demonstrated significant improvements compared to mainstream architectures: the Top-1 classification accuracy increased by up to 4.56% over traditional CNN frameworks (e.g., VGG16) and achieved an overall 3.02% enhancement compared to pure Transformer baselines (e.g., Hiera-T-224). Ablation studies further validated the effectiveness of the module designs: the introduction of the Local Feature Enhancement Module (LFEM) improved classification accuracy by 1.81% over the baseline, while the subsequent integration of the Global–Local Feature Fusion Module (GLFFM) expanded the cumulative improvement to 3.02% [13]. However, our previous research has only explored the establishment of AI technology model for fly species identification, which can identify limited sarcosaphagous flies and cannot achieve the ready recognition of images at the end of convenient movement.

In this study, we aim to achieve innovation from theoretical breakthroughs to practical implementation through the solution of “database expansion—lightweight tools—technology integration”, and transform cutting-edge algorithms into productivity tools that can directly serve the detection of cases. We extend the database of sarcosaphagous fly species that can be identified under this architectural model to make it more suitable for forensic entomology, and on the other hand, we developed a WeChat Mini Program (WMP) based on this model to achieve convenient operation on mobile devices. WMP is a lightweight application platform owned by Tencent and is widely popular in China for its convenience and accessibility. And initially construct a molecular species rapid examination system based on DNA Barcoding technology as an aid for possible identification difficulties and suspicious results.

## 2. Materials and Methods

### 2.1. Collection of Samples and Images

#### 2.1.1. Collection of Samples

We used fish carcasses as bait to capture necrophilous flies. Captured flies were fixed in 70% alcohol and photographed for subsequent AI-based identification.

#### 2.1.2. Collection of Images

A randomly selected sample was placed under a XTL-24A computerized stereomicroscope, manufactured by Shanghai Puda Optical Instrument Co., Ltd. (Shanghai, China), and photographed sequentially. Positions of the samples included prone, supine, and lateral positions, as well as several random positions. Images of the samples under the microscope were captured using ToupTek ToupView 3.7 software.

Another portion of the sample was photographed using six different smartphone models, including three mobile phone models under the current mainstream brands in China, Vivo X27, iQOO Neo 5 (Vivo Mobile Communication Co., Ltd., Dongguan, China), Redmi 13C (Xiaomi Corporation, Beijing, China), Nova 8 (Huawei Technologies Co., Ltd., Shenzhen, China) and two iOS-based iPhone SE, iPhone 13 (Apple Inc., Cupertino, CA, USA). In order to exclude the influence of the background and more in line with the forensic science practice, a white paper background was employed for sample photography to ensure consistent lighting and reduce visual interference. The flies were placed flat on the paper and photographed at random angles on a cloudy day at midday under natural outdoor lighting. The camera parameters of the mobile phone were set to default values and the camera was kept at a distance of about 10 cm from the subject.

At the end of the image collection, the samples were artificially classified by morphology, and the species were determined by combining the identification results of insect molecular species in the next step. These images were eventually incorporated into the insect image AI recognition training database.

### 2.2. Insect Species Determination

#### 2.2.1. DNA Extraction of Samples

Insects were classified morphologically and their DNA was extracted using the Neasy^®^ Blood &Tissue Kit (QIAGEN, Hilden, Germany) according to the recommended procedure of the kit combined with the summary of the previous method in our laboratory [14]. The extracted DNA was detected by 2% agarose gel electrophoresis and stored at 4 °C for use.

#### 2.2.2. Polymerase Chain Reaction

All samples were amplified using the COI (Cytochrome coxidase subunit I) gene universal primers LCO1490: 5′-GGTCAACAAATCATAAAGATATTGG-3′, HCO2198: 5′-TAAACTTCAGGGTGACCAAAAAATCA-3′ amplification of all samples [15]. The PCR reaction system was 40 μL, including 2 μL of DNA template, 2 μL each of 10 μL upstream and downstream primers, 20 μL of MonAmpTM 2× Taq mix Pro, and 14 μL of ddH_2_O to make up the system. The reaction procedure was as follows: pre-denaturation at 94 °C for 5 min, denaturation at 94 °C for 30 s, annealing at 50 °C for 45 s, extension at 72 °C for 40 s, a total of 35 cycles, and finally extension at 72 °C for 10 min. The PCR products were detected by 2% gel electrophoresis, and then sent to Taihe Biotechnology Co., Ltd. (Beijing, China) for bi-directional Sanger sequencing for the purpose of ensuring the accuracy of the sequencing results.

#### 2.2.3. Sanger Sequencing

Sanger sequencing was performed using the BigDye Terminator v3.1 Cycle Sequencing Kit. The sequencing reaction mixture was prepared in the following proportions (on ice): 0.5 μL Sequencing RR100, 2 μL 5× Sequence Buffer, 1 μL (3.2 pmol/μL) of sequencing primer, 1.5 μL of template (PCR product or plasmid), and deionised water to make up to a total volume of 10 μL. Sequencing reactions were performed in the following conditions: pre-denaturation for 2 min at 96 °C, followed by 34 cycles (10 s at 96 °C, 5 s at 50 °C, and 4 min at 60 °C). minutes, followed by 34 cycles (96 °C for 10 s, 50 °C for 5 s, 60 °C for 4 min), and finally maintained at 4 °C. After the sequencing reaction was completed, the product was purified by ethanol precipitation to remove unbound dye terminators and impurities. The purified product was analysed by electrophoresis on an ABI 3730XL sequencer to obtain high-quality sequencing data (Waltham, MA, USA).

#### 2.2.4. Molecular Species Identification

The forward and reverse sequences from sequencing were spliced and corrected by the software Snapgene.8.0.1 to ensure sequence accuracy and integrity. The final generated spliced sequences were used for subsequent analyses. The spliced and corrected sequences were aligned with known species sequences in the GenBank database by the BLAST (https://blast.ncbi.nlm.nih.gov/Blast.cgi, accessed on 8 May 2025) tool of National Center for Biotechnology Information (NCBI). Among the comparison results, the entries with 100% similarity (Percent Identity) and coverage (Query Coverage) were selected as the basis for identification. For samples with sequences having similarity of 95–100% in the BLAST results, the software MEGA.11 was used to determine the species through multiple sequence alignment and phylogenetic tree construction. Finally, the results were compared with morphological observations to ensure the accuracy of species identification.

### 2.3. Dataset Expansion and Model Training

#### 2.3.1. Data Set Preparation

For the screening of the captured fly images, all images were automatically pre-processed by a computer program, including cropping, scaling to a uniform size (320 px × 320 px) and normalisation to fit the input requirements of the deep learning model. Using the t torchvision.transforms.Compose of PyTorch 2.3.0, we performed geometric transformations such as random horizontal flip, rotation, affine transformation and colour perturbation with brightness, contrast and saturation adjustments on the images to ensure the diversity of the generated samples, thus avoiding the redundancy of information in the high sample categories due to oversampling, and at the same time mitigating the risk of underfitting in the low sample categories. The model randomly selects 90% of the images as the training set, 5% as the validation set, and 5% as the test set.

#### 2.3.2. Model Architecture Selection

The newly added fly image dataset prepared in the previous step was placed on a framework called GLB-ViT for the fine-grained image classification task of flies, efficiently integrating the global and local information at different levels in the image, which was developed by our group earlier. The GLB-ViT architecture addresses the need for the hierarchical recognition of morphological features of Diptera insects. The baseline network of GLB-ViT is the Hiera-Small-224 model (Hierarchical Efficient Representation Architecture) [16]. As shown in Figure 1 it consists of three key components: the Hiera Encoder block for global feature extraction; the Local Feature Extraction Module (LFEM) for local information mining, where the CBAM module enhances the extraction of local features by adaptively emphasising important channels and spatial regions; and the Global and Local Feature Fusion Module (GLFFM) to integrate both types of features. After fusion, an MLP outputs classification results [13]. This framework’s effective combination of global and local features offers a robust solution for forensic fly identification.

#### 2.3.3. Training Strategies

We followed the same training protocol as Hiera. Input images were randomly cropped to a size of 224 × 224. Cross-entropy was employed as the loss function, and label smoothing was used. The positional embedding learning rate decay ratio was set to 0.5. The AdamW optimiser with a momentum of 0.9 was utilised. Training lasted for 150 epochs, with a batch size of 32 and a learning rate of 0.0002. During validation, validation images were centre-cropped.

### 2.4. Building and Testing the WMP

#### 2.4.1. System Architecture

The system adopts a client–server collaborative architecture to build a complete image acquisition → recognition → feedback business closed loop. The mobile terminal provides an intuitive interactive interface that supports two input modes: users can either upload historical specimen images from the local gallery or call the camera to take real-time shots. After the image is uploaded, it automatically performs intelligent cropping, focuses on the main area of the insect and standardises it to 224 × 224 pixel-input specifications. The server side deploys an optimised lightweight recognition engine, which receives the image data through a RESTful API (Representational State Transfer) interface and returns Top-1 species recognition results and confidence assessment.

#### 2.4.2. Model Deployment

This system adopts a hierarchical deployment strategy to effectively balance the real-time performance of the mobile terminal and the computational accuracy requirements of the cloud. Through industry-standard model quantisation techniques, the original model is transformed into a lightweight model, which significantly reduces the resource consumption of terminal equipment. The deployment scheme contains two collaborative modules: (1) mobile real-time preview: deploying the lightweight model to achieve real-time feedback on shooting, assisting the user to quickly adjust the shooting angle and focus range; and (2) cloud high-precision identification: triggering the calculation of the complete model after the user confirms the submission to ensure the reliability of the identification in complex scenarios. This strategy takes into account both efficiency and precision, and achieves a smooth interactive experience on mainstream mobile devices.

#### 2.4.3. Tests of AI Auto-Recognition Effectiveness

After uploading the test set images to the WMP’s back-end server, they are recognised by the trained deep learning model while monitoring the server operation status. Each identification result is recorded, and based on the identification results and the comparison with molecular biology and artificial morphological typing, a confusion matrix is constructed, and the indexes such as recall rate, precision rate F1 score are calculated. These indicators are based on four basic indicators in the confusion matrix: true positives (TP), false positives (FP), false negatives (FN), and true negatives (TN).$$Precision\;rate=\frac{TP}{TP+FP}$$$$Recall\;rate=\frac{TP}{TP+FN}$$$$F1\;Score=2\cdot \frac{Precision\;rate\cdot Recall\;rate}{Precision\;rate+Recall\;rate}$$

In addition, we calculated the probability of species A being misclassified as species B and the probability of species B being misclassified as species A to evaluate which species are most likely to be confused with each other. Specifically, the Symmetric Error Rate (SER) is defined as the average of the probabilities of two samples being misclassified as each other.$$SER=\frac{1}{2}\left(\frac{N_{A\to B}}{N_{A}}+\frac{N_{B\to A}}{N_{B}}\right)$$

N_A→B_ and N_B→A_ denote the number of samples that were actually specie A and B, respectively, but were incorrectly predicted to be another specie.

The next step is the construction of a rapid molecular species identification system for flies that are easily misidentified by the WMP.

### 2.5. Establishment of a Molecular Species Identification System

#### 2.5.1. Selection of Primers

For species that are difficult to identify and differentiate by the WMP, their DNA was amplified using two pairs of primers that can effectively differentiate between different fly species (in the case of species that were not able to be captured in this study, stale DNA samples that had been typed in the previous period by our group were used). After extensive screening and combining the results of AI identification, two pairs of primers were finally used:

Primer A is a fly cox1 primer designed by Liu [17], and the length of the amplified fragment was 272 bp. The sequences were F: 5′-CAGATCGAAATTTAAATACTTC-3′; R: 5′-GTATCAACATCTATTCCTAC-3′. The primers were annealed at 50 °C. Primer B (C1-J-2495/C1-N-2800) amplification site is located in mtDNA [18], and the length of the amplified fragment is 278 bp. The length of the amplified fragment was 278 bp, the primers were annealed at 55 °C, and the sequences were as follows: C1-J-2495: 5′-CAGCTACTTTATGAGCTTTAGG-3′, C1-N-2800: 5′-CATTTCAAGCTGTGTAAGCATC-3′.

#### 2.5.2. Polymerase Chain Reaction

The PCR system was set to 20 μL, including 1 μL of DNA template, 1 μL each of upstream and downstream primers, 1 μL of 20× EvaGreen saturated fluorescent dye, 10 μL of MonAmpTM 2× Taq mix Pro, and 6 μL of ddH_2_O to make up the system. In the same PCR system, ddH_2_O was also used in place of DNA samples to serve as a blank control. PCR reaction conditions were pre-denaturation at 94 °C for 5 min, denaturation at 94 °C for 30 s, annealing at the annealing temperature corresponding to each primer for 40 s, and extension at 72 °C for 40 s for a total of 35 cycles, and final extension at 72 °C for 10 min.

#### 2.5.3. High Resolution Melting Analysis

The PCR products were analysed on the Rotor-Gene Q Fluorescence PCR Analyser (Hilden, Germany). The conditions were set as follows: ramp from 60 degrees to 95 degrees, increasing by 0.1 degree per step. Wait for 90 s of pre-melt conditioning at the first step, and for 2 Optimise the gain before melting on all tubes. Select the gain that gives the highest fluorescence value below 95 degrees. Compare the final melting curves obtained to the amplified curves of samples that are clearly of that species.

## 3. Results

### 3.1. New Insect Species and Image Count

We collected and stored a total of 3189 new original images in this study. As shown in Figure 2, each subject was photographed with images in prone, supine, and lateral positions, in addition to several random posture images.

In order to ensure the accuracy of new species, the samples were confirmed by both morphological and molecular biological identification. The newly acquired images included seven necrophilous fly species, of which four species were newly added: *Sarcophaga ruficornis* (Fabricius, 1794) (Diptera: Sarcophagidae), *Chrysomya megacephala* (Fabricius, 1794) (Diptera: Calliphoridae), *Chrysomya rufifacies* (Macquart, 1843) (Diptera: Calliphoridae), *Synthesiomyia nudiseta* (van der Wulp, 1883) (Diptera: Muscidae). The entry images of each fly species in the database of this model are shown in Table 1.

### 3.2. Status of AI Training

After oversampling the original images obtained, in order to ensure a balanced distribution of each category and to cover the dorsal (prone position), ventral (supine position), lateral (lateral recumbent position) and other morphological features of insects, the number of images of each species was unified to between 1010 and 1200, and finally 11,331 high-definition images were obtained, which contained a total of 10 species of common Diptera insects. In total, 90% of the images were taken as the training set, while the validation set and the test set took 5% each, and each type of dataset existed independently, as shown in Table 2.

### 3.3. Establishment of WMP

Based on the trained model, we successfully developed a Mini Program for fly species identification based on image recognition, named Forensic-Insect Identification 1.0 (FII 1.0), which supports both iOS and Android systems. The user interface of FII 1.0 is designed to be simple and intuitive, which is easy for users to operate. The main functions of FII 1.0 include real-time photo identification, local image uploading and identification result display. As shown in Figure 3, WMP calls the device camera or photo album to obtain the fly image, and transmits the image to the back-end server for real-time recognition, and the recognition result will be returned to the user through the WMP interface to display the species name of the fly.

### 3.4. AI System Testing

To evaluate the Mini Program’s performance in fly recognition, we tested it using an independent test set of 567 images, and correctly recognised species from 533 images with a model accuracy of 94.00%, using molecular biology and manual morphological identification results as a reference criterion, and Figure 4 visualises the recognition results of the images used for testing for each species.

From the detailed output of the terminal, the correct recognition rates of different fly species varied, all above 80%. *L. cuprina* had the highest correct recognition rate of 100%, while *S. peregrina* had the lowest at 80.70%. In addition, we also counted the overall recognition rates among different genera and families in Table 3.

At the family level, the present AI model correctly identified 261 images of Sarcophagidae (93.21% correct), 226 of Calliphoridae (96.58% correct), and 45 of Muscidae (86.79% correct). At the genus level, the present AI model correctly identified *Sarcophaga* 261 (93.21% correct), *Chrysomya* 172 (95.56% correct), *Lucilia* 54 (100% correct) and *Synthesiomyia* 45 (86.79% correct). To represent the generalisation of the model, a confusion matrix was used, along with computational precision, recall and F1 score as metrics.

The confusion matrix in Figure 5 provides a visual representation of the model’s classification results. Blue squares along the diagonal indicate correct classification, while squares above and below the diagonal indicate misclassification [19].

Precision, recall, and F1 scores for each fly species varied but were all greater than 80.00%, indicating a good overall performance. The F1 score combines recall and precision. Species such as *S. sericea* (99.16%), *L. cuprina* (99.08%), and *C. pinguis* (98.31%) achieved high F1 scores, suggesting a balance between the two metrics and excellent recognition performance. In contrast, *S. peregrina* (85.19%) had a relatively low F1 score. This species needs improvement in both recall and precision (Figure 6).

Figure 7 illustrates the SER among various pairs of fly species. This metric reflects the probability that any two fly species are misclassified as each other. Notably, significant variations exist in the SER across different fly–species pairs. Certain species pairs, for instance *S. dux* and *S. sericea* (exhibiting SER of 0.00%), as well as *C. pinguis* and *C. rufifacies* (with identical SER of 0.00%), demonstrate exceedingly low SER. This phenomenon provides strong evidence that the model showcases exceptional performance in differentiating these species pairs.

Conversely, species pairs including *C. megacephala* and *C. rufifacies* (registering SER of 4.17%) and *S. misera* and *S. peregrina* (with SER of 8.88%) present relatively high SER. To enhance the discrimination ability between these easily-confused species, we further developed a DNA barcoding system.

### 3.5. Establishment of the HRM Rapid Inspection System

PCR-HRM experiments were performed for the two groups of species most likely to be misidentified by the WMP, *S. misera* and *S. peregrina*, and *C. megacephala* and *C. rufifacies*, and as shown in Figure 8, all four species could be amplified and single peak melting curves were generated. The blank control resulted in negative findings.

The melting profile of the PCR products was obtained after repeating the experiment three times under the same conditions.

As shown in Table 4, Primer A formed melting curves with a Tm difference of at least greater than 0.55 °C each time *S. misera* and *S. peregrina* were amplified; the difference in Tm values was within 0.25 °C when compared with positive samples of known species.

As shown in Table 5, Primer B formed melting curves with a Tm difference of at least 1.15 °C each time it amplified the distinction between *C. megacephala* and *C. rufifacies*. When compared with positive samples of known species, the Tm values all differed within 0.20 °C.

## 4. Discussion

### 4.1. Performance of the Model

We tested the recognition performance of the WMP using a test set of images containing 567 samples independent of the model training set, and the accuracy reached 94.00%, which is basically the same as that of experienced forensic entomologists and NCBI BLAST based on Sanger sequencing. It initially indicates that the model has certain reliability and practical value in the field of fly species recognition for relevant practical applications.

Although the model achieved some results in fly species identification, the heat map of the confusion matrix provided shows that there are still two types of errors in the model: the first one is the confusion of close species, which is the main confusing category, for example, the misclassification rate between the species of *S. misera* and *S. sericea* in the same genus is 0.83%, and the misclassification rate between the species of the same family, *C. megacephala* and *C. rufifacies*, is 4.17%, which is the high misclassification rate among the close species. This high misclassification rate between similar species is mainly due to their high morphological similarity, which makes it difficult for the model to accurately capture those subtle but distinguishing features; secondly, it is a cross-family misclassification, in which *S. nudiseta* is particularly prominent. The fly can be misclassified as *S. dux*, *S. peregrina*, *C. pinguis*, *L. cuprina* and other species. On the one hand, this is due to the similarity of morphological characteristics among different fly species; on the other hand, as a new species in the model database, the sample size of the *S. nudiseta* is relatively small. The insufficient sample size limits the model’s ability to fully learn the features of the species, resulting in the difficulty in grasping its unique features, which in turn affects the recognition accuracy.

### 4.2. Comparative Analysis with Similar Models

In this study, the selected fly species with darker body colours were photographed by placing them on a white background. It fit the operational conditions of forensic science practice and effectively avoided the interference of complex backgrounds on the images, while enhancing the image contrast in the training and inference stages [20]. However, the insect samples of some species used in this study were all dead when the pictures were collected, and the colour of the insect corpses had changed to a certain extent, and the limbs and trunks were twisted or even absent. The change in and absence of appearance may be an important factor affecting the recognition rate [21], which makes identification more difficult. Follow-up studies need to consider how to overcome these sample deficiencies, such as exploring more scientific methods of preserving and photographing samples, or adopting more advanced image-processing techniques to repair and enhance the images of damaged samples, or developing image recognition AI models targeted at this type of sample.

The database of this model, updated by this study, was able to achieve preliminary discrimination of 10 sarcosaphagous fly species in four genera. In contrast, some other studies achieved high accuracy in genus-level classification but lacked in-depth classification within the same genus [19,20]. There were also studies covering multiple families and fly variants, yet they did not deal with the identification of different species within the same genus [22]. This suggests that this study is pioneering in the exploration of species-level classification, adding the ability to classify flies more accurately based on AI methods. Unlike two-dimensional objects, the fly carapace is a three-dimensional structure, and the feature points for morphological classification may be distributed in multiple planes. Currently, there are studies to distinguish necrophilous flies by decision tree methods, random forest algorithms combined with wing morphometric data [23,24,25], or identification of species by CNNs image learning of posterior stomata [26], which achieved certain better results, providing a path for its continuation in feature mining and model optimisation after this study, and thus expected to further improve the recognition accuracy.

At the application level, the WMP developed in this study has the advantage of being able to run on mobile phones. Unlike studies relying on professional cameras to take pictures, we used mobile phones and microscopes to acquire images, fitting more practical application scenarios. However, at present, the WMP has only completed the test of importing images from photo albums, and the stability of the on-site photo function has not yet been verified, so we need to focus on this aspect of the test work in the future to ensure the reliability of the WMP in practical use. On the other hand, users should ideally also require some training. Knowing basic entomological knowledge and the limitations of AI may help improve the effectiveness of AI methods in forensic medicine.

### 4.3. About the Molecular Rapid Test HRM

Morphological datasets of insects typically have a large number of classes and very fine-grained distributions, where phenotypic differences between species may be large and differences within species may be small. Such datasets are particularly challenging for open-set identification methods. While it is very difficult to overcome this challenge for all species using only phenotype-based identification, a combination of image-based deep learning and DNA barcoding techniques may help to solve this problem [27]. HRM techniques do not require specific marker labelling probes, they are simple, low-cost and high-throughput, and are capable of detecting differences in single nucleotides [28], and the species identification of flies can be completed within three hours after DNA extraction [29]. HRM technology is a convenient detection method, as it can achieve the rapid identification of insect species, which has led to its widespread interest in the field of forensic science [30,31].

We combined the HRM technique with the DNA barcoding technique, which can effectively distinguish two pairs of species that are easily confused by the AI model by examining the COI gene of flies. This shows that HRM technology has good advantages in the identification of necrophilous flies, especially in the rapid distinction of closely related species. Combined with WMP, HRM can quickly achieve the distinction of ten fly species with only two pairs of primers, reducing the time under the premise of ensuring the accuracy of forensic insect species identification.

## 5. Conclusions

By expanding the dataset, we optimised the GLB-ViT model. The optimised model can successfully identify images taken by mobile phones and microscopes of three families, four genera, and ten species of flies.

The WMP we developed initially applied the model to real-world forensic science scenarios and achieved an overall accuracy of 94.00%, which is comparable to the accuracy of species identification by forensic entomologists and specific sequencing methods. The DNA barcoding system we established can effectively assist in dealing with the difficulties and suspicious results that may occur in the model identification. The combination of the two provides new ideas and technical means for the practice of forensic entomology, and, to a certain extent, will alleviate the dilemmas faced by relying on traditional methods.

In order to further optimise the fly identification technology, the sample size of confusable species will be expanded, and new species will be added in the future. Meanwhile, we will optimise primer design and selection and construct a high-quality HRM-DNA barcode database. Ultimately, the AI model and molecular techniques complement each other, expanding the scope for practical applications of sarcosaphagous fly identification.

## Figures and Tables

**Figure 1 insects-16-00529-f001:**
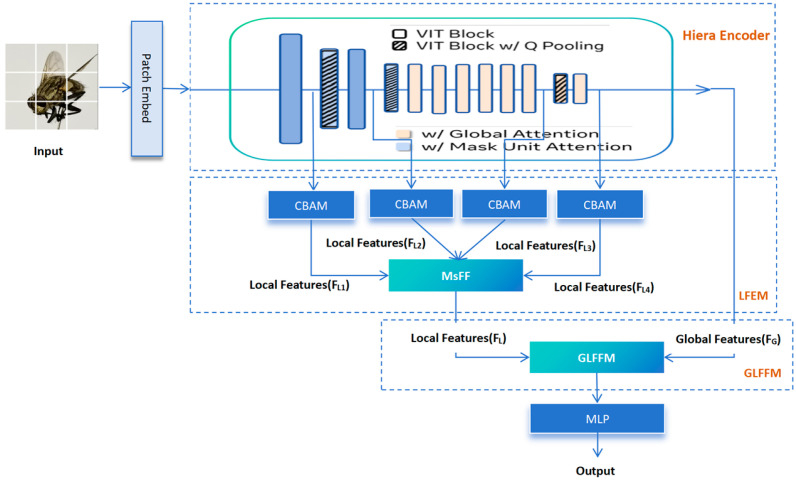
Schematic of GLB-ViT network architecture.

**Figure 2 insects-16-00529-f002:**
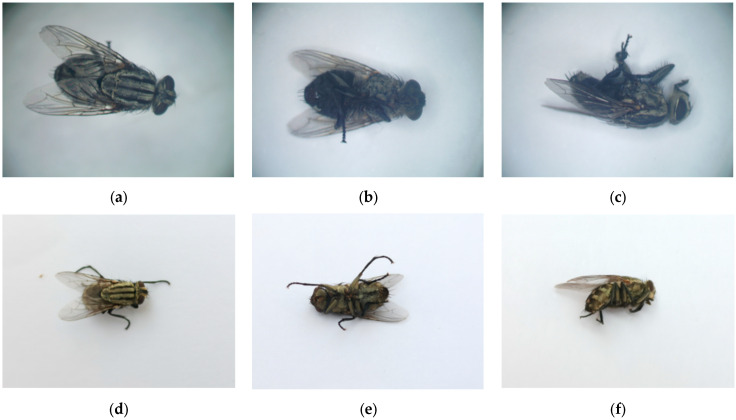
Exemplary images of the dataset As an example, (**a**–**c**) show microscopic images of *S. dux* in supine, prone, and lateral positions, respectively. (**d**–**f**) present mobile phone-captured images of pairs of *S. dux* in supine, prone, and lateral positions, respectively.

**Figure 3 insects-16-00529-f003:**
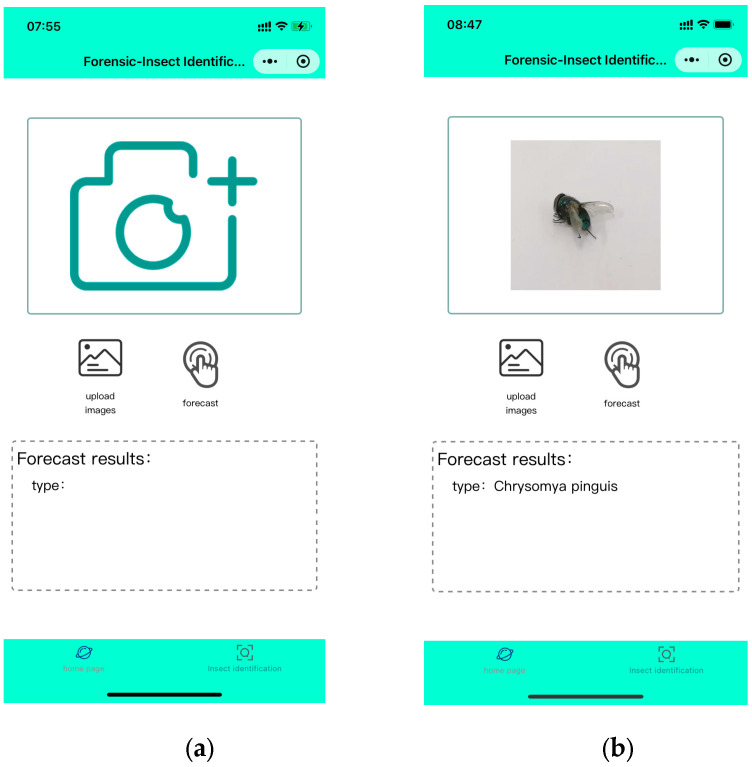
Developed mobile applications used on smartphones: (**a**) shows the Mini Program, the main interface, which gives users the option of taking a picture or selecting an image from their gallery; (**b**) shows the detection interface, displaying images of the samples taken or uploaded by the user, as well as the recognition results.

**Figure 4 insects-16-00529-f004:**
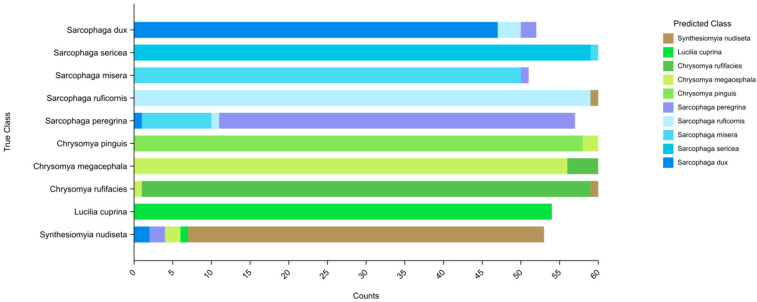
Stacked histogram of image recognition results for each species. The vertical axis represents the real species of the sample used for testing and the horizontal axis is the number of images.

**Figure 5 insects-16-00529-f005:**
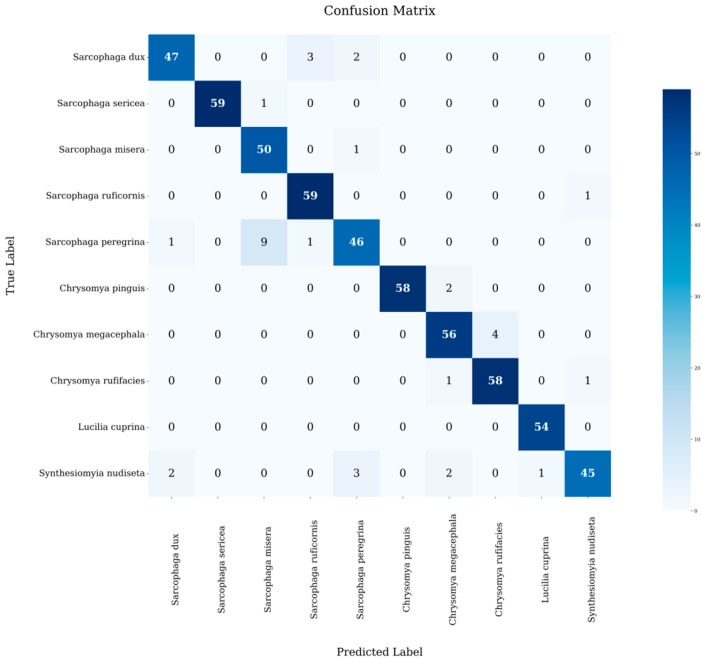
Heat map of the confusion matrix.

**Figure 6 insects-16-00529-f006:**
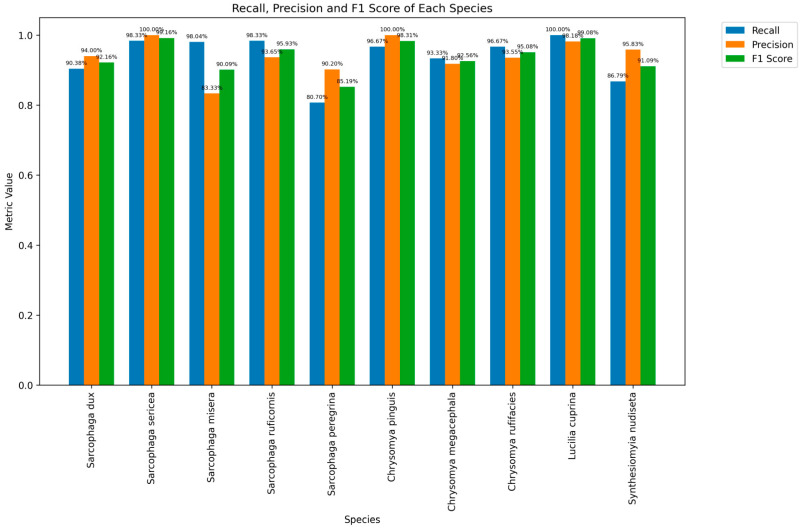
Histograms of precision, recall and F1 scores for each species.

**Figure 7 insects-16-00529-f007:**
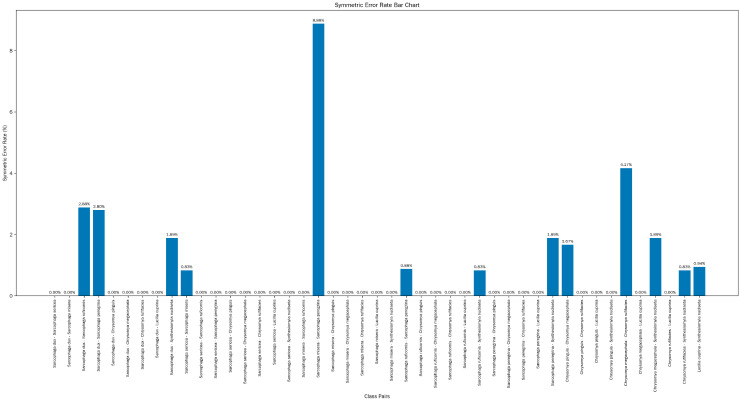
Mean per-class error of fly species pairs.

**Figure 8 insects-16-00529-f008:**
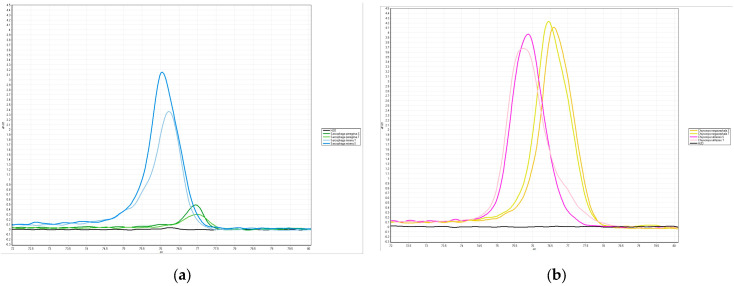
HRM Curves of Samples: (**a**) the HRM curves obtained by Primer A amplification of *S. misera* and *S. peregrina*; (**b**) the HRM curves obtained by Primer B amplification of *C. megacephala* and *C. rufifacies*. The “.S” indicates the positive control (standard), and “.T” indicates the sample used for testing.

**Table 1 insects-16-00529-t001:** Number of original images of flies added to each fly species.

Species	Initial	Increment	Total
*Sarcophaga dux*	372	296	668
*Sarcophaga sericea*	405	0	405
*Sarcophaga misera*	505	0	505
*Sarcophaga peregrina*	72	502	574
*Chrysomya pinguis*	2029	393	2422
*Lucilia cuprina*	360	0	360
*Sarcophaga ruficornis **	0	120	120
*Chrysomya megacephala **	0	731	731
*Chrysomya rufifacies **	0	796	796
*Synthesiomyia nudiseta **	0	351	351
Total	3743	3189	6932

Species newly added by this study are marked with *.

**Table 2 insects-16-00529-t002:** Composition and distribution of the data set.

Species	Training	Verifying	Testing	Total
*Sarcophaga dux*	936	52	52	1040
*Sarcophaga sericea*	1080	60	60	1200
*Sarcophaga misera*	909	50	51	1010
*Sarcophaga ruficornis*	1080	60	60	1200
*Sarcophaga peregrina*	1033	58	57	1148
*Chrysomya pinguis*	1080	60	60	1200
*Chrysomya megacephala*	1080	60	60	1200
*Chrysomya rufifacies*	1080	60	60	1200
*Lucilia cuprina*	972	54	54	1080
*Synthesiomyia nudiseta*	947	53	53	1053
Total	10,197	567	567	11,331

**Table 3 insects-16-00529-t003:** Overall recognition accuracy at different taxonomic levels.

Families	Genera	Species	Accuracy (Species)	Accuracy (Genus)	Accuracy (Family)
Sarcophagidae	*Sarcophaga*	*Sarcophaga dux*	90.38%	93.21%	93.21%
		*Sarcophaga misera*	98.33%		
		*Sarcophaga sericea*	98.04%		
		*Sarcophaga ruficornis*	98.33%		
		*Sarcophaga peregrina*	80.70%		
Calliphoridae	*Chrysomya*	*Chrysomya pinguis*	96.67%	95.56%	99.57%
		*Chrysomya megacephala*	93.33%		
		*Chrysomya rufifacies*	96.67%		
	*Lucilia*	*Lucilia cuprina*	100.00%	100.00%	
Muscidae	*Synthesiomyia*	*Synthesiomyia nudiseta*	86.79%	86.79%	86.79%

**Table 4 insects-16-00529-t004:** Melting temperatures of *S. misera* and *S. peregrina*.

Species	*Sarcophaga misera*		*Sarcophaga peregrina*	
Sample	Standard	Testing	Standard	Testing
Tm1 (°C)	76.02	76.23	76.78	76.87
Tm2 (°C)	76.05	76.22	76.95	76.98
Tm3 (°C)	76.00	76.25	76.65	76.80

**Table 5 insects-16-00529-t005:** Melting temperatures of *C. megacephala* and *C. rufifacies*.

Species	*Chrysomya megacephala*		*Chrysomya rufifacies*	
Samples	Standard	Testing	Standard	Testing
Tm1 (°C)	76.85	76.65	75.53	75.48
Tm2 (°C)	76.20	76.25	75.75	75.62
Tm3 (°C)	76.45	76.60	75.88	75.75

## Data Availability

The original contributions presented in this study are included in the article. Further inquiries can be directed to the corresponding authors.

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
