# Peer review of "Preliminary Development of Global–Local Balanced Vision Transformer Deep Learning with DNA Barcoding for Automated Identification and Validation of Forensic Sarcosaphagous Flies"

_insects, 2025, doi:10.3390/insects16050529_

Round 1
Reviewer 1 Report
Comments and Suggestions for Authors
Title: Preliminary Development of GLB-ViT Deep Learning with DNA Barcoding for Automated Identification and Validation of Forensic Sarcosaphagous Flies
Overview and General Recommendation:
The article applies deep learning techniques to insect recognition, which aligns well with the journal's theme and demonstrates strong practical value. However, the following shortcomings are noted and could be improved:
- The manuscript mentions previous work related to GLB-ViT. Please cite the earlier work in the main text and provide a brief introduction to it.
2.The manuscript uses the Hiera-Small-224 model. Please briefly describe the FPN (Feature Pyramid Network) structure within the model and how feature fusion is achieved. It is recommended to include a schematic diagram or relevant formulas to support the explanation. If related work exists, it can be cited directly. Additionally, please include the evaluation metric formula used in the accuracy assessment.
3.The results table showing recognition accuracy lacks a comparison with other methods. Please add at least one baseline method to demonstrate the superiority of the proposed approach.
4.Please review the entire manuscript to ensure proper formatting (e.g., avoid large blank areas) and ensure the narrative is complete (e.g., line 363).
5.The figures in the manuscript are not clear, and not all high-resolution images were provided in the supplementary files. Please supplement them accordingly.
6.The manuscript lacks a clear description of its innovations. Please highlight the novel contributions to enhance the overall quality of the paper.
Please address each of these points in your revision. The paper has the potential to make a valuable contribution to the field, but it requires these revisions to meet the publication standards.
Comments on the Quality of English LanguageThe paper is of high English quality, with smooth narration and no obvious grammar errors. It basically meets the requirements for academic publication.
Reviewer 2 Report
Comments and Suggestions for Authors
The paper is well written, with good structure and clear English throughout. The technical terminology is appropriate for the field. However, I would like to highlight a few points:
-The only new element is a custom dataset; the GLB-ViT model is off-the-shelf and not novel. The paper lacks sufficient technical detail or justification for its selection.
-There is no comparison with standard models (e.g., CNNs, ResNet, EfficientNet, or other ViT variants). It would be useful to see how these models perform in comparison.
-The paper does not include ablation studies (e.g., model performance with and without DNA barcoding, or across different architectures).
Comments on the Quality of English Language
A few areas could be improved for clarity and fluency:
e.g. this sentence "the development of related applications also provides" in page 2 is repeated
Reviewer 3 Report
Comments and Suggestions for Authors
The paper is the evolution of a previous work reported in the reference [13], where the GLB-Vit model has been presented with more details and applied to a subset of flies.
The paper here is interesting and have a good potential for forensic applications.
Just few questions:
1) a part from the color of the flies, did the alcohol solution change anything else?
2)Data on Table n.2: The number of validation and test samples seems to be less than 15% mentioned in the text above the table, there's a reason for that?
3)The recognition rate of Sarcophaga peregrina is 80%, it is due to the quality of the samples?
Comments:
Line 62: it seems there's a repetition of 'postmortem';
Line 74 : The method can help when the lack of experience of the operator is an issue, but in case of 'false positives' the lack of experience is more critical. The operator itself should be trained enough before using any AI method.
Round 2
Reviewer 2 Report
Comments and Suggestions for Authors
Thank you to the authors for the efforts tom improve the manuscript. However, I notice that a comparison with other relevant architectures is still missing. Such comparisons is essential to justify the choice of the proposed model, ResNet, ResNext e.g. A figure about the proposed pipleline is needed as well.
